# RAID: A Benchmark Dataset for Testing the Adversarial Robustness of AI-Generated Image Detectors

## Abstract

AI-generated images have reached a quality level at which humans are incapable of reliably distinguishing them from real images. To counteract the inherent risk of fraud and disinformation, the detection of AI-generated images is a pressing challenge and an active research topic. While many of the presented methods claim to achieve high detection accuracy, they are usually evaluated under idealized conditions. In particular, the *adversarial robustness* is often neglected, potentially due to a lack of awareness or the substantial effort required to conduct a comprehensive robustness analysis. In this work, we tackle this problem by providing a simpler means to assess the robustness of AI-generated image detectors. We present **RAID** (**R**obust evaluation of **AI**-generated image **D**etectors), a benchmark dataset of 72k diverse and highly transferable adversarial examples. The proposed dataset is created by running attacks against an ensemble of seven state-of-the-art detectors and images generated by four different text-to-image models. Extensive experiments show that our methodology generates adversarial images that transfer with a high success rate to unseen detectors, which can be used to quickly provide an approximate yet still reliable estimate of a detector's adversarial robustness. Our findings indicate that current state-of-the-art AI-generated image detectors can be easily deceived by adversarial examples, highlighting the critical need for the development of more robust methods.

## 1 Introduction

In recent years, generative artificial intelligence has evolved from a mere research topic to a vast collection of commonly available tools. While the inception of large language models, most notably ChatGPT, has been most transformative for our everyday life, the evolution of generative image modeling has drastically shifted our understanding of visual media. This development was initiated by the discovery of diffusion models (Sohl-Dickstein et al., 2015), which utilize an iterative noising and denoising process (Ho et al., 2020) to learn the distribution of natural images. Later work (Rombach et al., 2022) improved this process by performing the generation process in the compressed latent space of a pre-trained variational autoencoder (Kingma & Welling, 2014) that essentially reduces computational overhead and preserves semantic information while discarding high-frequency noise, in addition to introducing flexible conditional generation with the use of cross-attention layers. This rapid development, while improving computer vision tasks such as image upsampling (Wu et al., 2024) and dataset augmentation (Azizi et al., 2023), poses a considerable risk of nefarious misuse leading to the spread of misinformation, privacy violation, and identity theft (Yang et al., 2024; Ricker et al., 2024a). This underscores the urgent need for detection methods that generalize and keep up with the ever-evolving image generation technology while maintaining robustness to adversarial attempts to evade detection.

To mitigate the harmful consequences of AI-generated images (AIGIs), a variety of detection approaches have been proposed in the literature (Wang et al., 2020; Guo et al., 2022; Marra et al., 2019; Ojha et al., 2023; Chen et al., 2024a; Baraldi et al., 2024; Wang et al., 2023). Based on the reported results of near perfect accuracy, it seems the problem of detecting AIGIs is already solved. However, evaluations are typically conducted within an idealized lab setting that does not consider real-world risks. One major factor is the effect of common processing operations, such as resizing

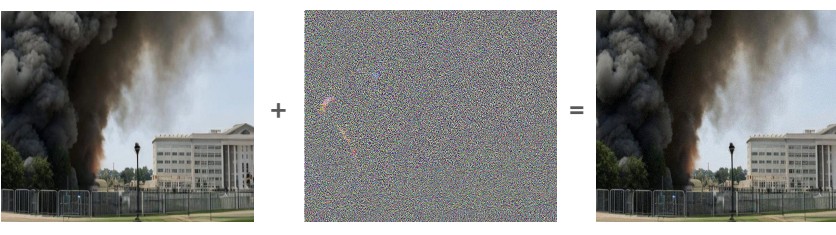

Figure 1: Adversarial attack on the commercially available detector provided by *sightengine*. Successful evasion is shown by adding adversarial noise computed on the detector by Corvi et al. (2023). We perform an extended evaluation on commercial detectors in Appendix C.

or compression, which have already been shown to have drastic effects on the performance of AIGI detectors (Gragnaniello et al., 2021; Xu et al., 2024; Cocchi et al., 2023).

An important factor, which the majority of existing work has neglected, is the *adversarial robustness* of detectors. Take, for instance Figure 1, a synthetic image generated and spread across social media outlets (Oremus et al., 2023) that, despite being quickly debunked as fake, still had an impact on the stock market. We can detect such an image as being AI-generated with an off-the-shelf detector, but when we modify the image using carefully designed adversarial perturbations, it can evade the detection of said detector and others, with the effectiveness increasing for those that share a similar architecture (Mavali et al., 2024). The adversarial robustness is often not investigated in works proposing synthetic image detectors, partially due to the significant effort required to generate adversarial examples. Due to many attack algorithms and hyperparameters and the required technical knowledge, conducting a comprehensive robustness analysis is not straightforward.

Existing work (Mavali et al., 2024) unveils this failure to show robustness in white-box scenarios where the malicious actor has access to the architecture and training parameters of the detector, and also in black-box scenarios where the attacker's knowledge is limited. However, we note that the attacks used for the evaluation remain restricted in using techniques that increase their success and transferability. In this work, we extend this concept to bridge this evaluation gap by providing a standard and effective means to assess the adversarial robustness of AIGI detectors. In particular, we propose **RAID** (**R**obust evaluation of **AI**-generated image **D**etectors), a large-scale benchmark dataset of *diverse* and *transferable* adversarial examples created using an ensemble of state-of-the-art detectors that employ different architectures. As we experimentally demonstrate, testing the detection performance on RAID provides a solid estimate of the adversarial robustness of a detector. Our benchmark on seven recently proposed detectors shows that the current landscape of AIGI detection is not yet expansive nor reliable for widespread adoption in the real world, without properly ensuring adversarial robustness to evasion attacks.

**Contributions.** In summary, we make the following contributions:

- We create RAID, the first benchmark dataset of transferable adversarial synthetic images, constructed using highly transferable attacks, to standardize testing the adversarial robustness of state-of-the-art synthetic image detectors.
- We conduct a large-scale study showing that adversarial perturbations transfer across several state-of-the-art synthetic image detectors.
- We show that the transferability of adversarial perturbations increases when we using an ensemble adversarial attack, with comparable results to a white-box attack.

## 2 THE RAID BENCHMARK

This section describes how we constructed our datset of transferable adversarial examples. An overview of how we created RAID is given in Figure 2.

### 2.1 SOURCE DATASET

RAID is built upon the $D^3$ dataset (Baraldi et al., 2024). In total, $D^3$ consists of 11.5M images. It is constructed from 2.3M real images taken from the LAION-400M (Schuhmann et al., 2021) dataset.

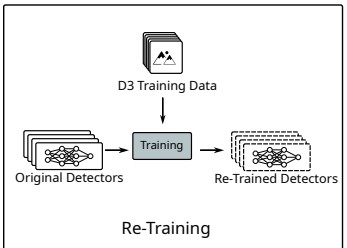 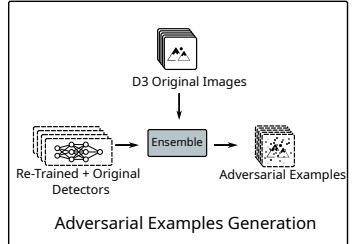 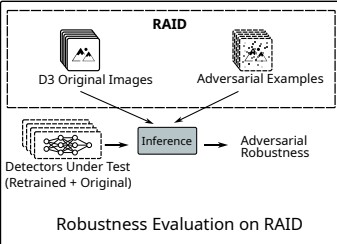

Figure 2: A diagram of the three experimental pipelines used in the paper for generating the RAID dataset. Left: We keep the top three performing detectors intact while we re-train the rest. Middle: We generate adversarial examples from adversarial attacks on an ensemble of detectors. Right: We evaluate the baseline detectors on the RAID images.

Using the corresponding caption as prompts, synthetic images are generated from four open-source text-to-image models: Stable Diffusion v1.4 (Rombach et al., 2022), Stable Diffusion v2.1 (Rombach et al., 2022), Stable Diffusion XL (Podell et al., 2023), and DeepFloyd IF (at StabilityAI, 2023). For details on data selection and prompt engineering, we refer to the original publication (Baraldi et al., 2024). It should be noted that each generated image is post-processed such that the image format and compression strength match that of the real distribution present in the corresponding real image. This not only reduces the risk of unwanted biases between real and generated images but also makes the dataset significantly more challenging for the detection task.

We built our dataset using $D^3$ in two phases. First, to re-train the detectors used to compute the adversarial examples, we take the training subset of $D^3$ comprising 2,311,429 real and 9,245,716 generated images. As we show in Section 3.2, re-training helps to ensure a sufficient detection performance on the original images and, subsequently, the generation of effective adversarial examples. Second, we use the same procedure as in (Baraldi et al., 2024) to construct the actual RAID dataset to generate synthetic images based on 4,800 new real images. For each of the resulting 24,000 images (*i.e.*, the real images and the synthetic images from four generators), we create matching adversarial examples using the attack presented in Section 2.2 for each $\epsilon$. Thus, our proposed benchmark dataset consists of 72,000 adversarial examples – 24,000 adversarial examples for each attack parameter $\epsilon$ ($\frac{8}{255}$, $\frac{16}{255}$, and $\frac{32}{255}$) – in addition to original images, for a total of 96,000 images.

## 2.2 CRAFTING ADVERSARIAL PERTURBATIONS

**Adversarial Examples Optimization.** Given an input $x \in [0,1]^d$ and a victim model with parameters $\theta$, adversarial examples (Biggio et al., 2013; Szegedy et al., 2014) can be crafted with evasion attacks, which aim to solve the following optimization problem to compute the adversarial perturbation $\delta \in \mathbb{R}^d$:

$$\arg\min_{\delta:\|\delta\|_\infty \leq \epsilon} \mathcal{L}(x + \delta, \theta) \ , \tag{1}$$

where $\epsilon$ is the applied bound on the perturbation size, and $\mathcal{L}$ is a loss function encoding the attacker's objective (*i.e.*, a misclassification).

The above optimization problem is commonly solved with gradient-based techniques, which require access to the model's parameters and gradients. Despite this approach working well in this white-box setting, it is often not possible to get full access to the model's internals. Nevertheless, it has been shown that the adversarial examples produced against a model can still be effective on different models, although this phenomenon, called *transferability*, is weaker across different model architectures. To increase transferability, previous works proposed to simultaneously attack an ensemble of models (Liu et al., 2017; Dong et al., 2018). We thus leverage an ensemble attack by extending Eq. 1 to a set of $M$ models:

$$\arg\min_{\delta:\|\delta\|_\infty \leq \epsilon} \mathcal{L}(x + \delta, \theta_1, \theta_2, \ldots, \theta_M) \ . \tag{2}$$

Among several possible choices, we define the loss function as:

$$\mathcal{L}(x + \delta, \theta_1, \theta_2, \ldots, \theta_M) = \frac{1}{M} \sum_{m=1}^{M} CE(l_{\theta_m}(x), y_t) \ , \tag{3}$$

where $CE$ is the cross-entropy loss, and $l_{\boldsymbol{\theta}_m}$ is the output logit of the $m$-th model and $y_t$ is the target class – *i.e.*, the output label desired by the attacker.

**Detection Ensemble.** The key requirement for our benchmark is that the adversarial examples are *transferable* (*i.e.*, they are effective against unseen detectors). To achieve high transferability, we use an ensemble of a diverse set of seven different detectors:

**Wang et al. (2020)** In this seminal work, the authors show that a ResNet-50 (He et al., 2016) trained on real and generated images from ProGAN (Karras et al., 2018) is sufficient to successfully detect images from a variety of other GANs. During training, extensive data augmentation is used to account for different image processing and increase generalization.

**Corvi et al. (2023)** This architecture is adapted from (Gragnaniello et al., 2021) and is a modification of the detector by Wang et al. (2020). It uses the same backbone, but avoids downsampling in the first layer to preserve high-frequency features and uses stronger augmentation.

**Ojha et al. (2023)** Unlike previous work, this detector uses a pre-trained vision foundation model (CLIP (Radford et al., 2021)) as a feature extractor to avoid overfitting on a particular class of generated images and, thus, improve generalization. To obtain a final classification, a single linear layer is added and trained on top of the 768-dimensional feature vector.

**Koutlis & Papadopoulos (2024)** This approach is similar to that presented by Ojha et al. (2023), but additionally uses intermediate encoder-blocks of CLIP. The resulting features are weighted using a learnable projection network, followed by a classification head.

**Cavia et al. (2024)** Instead of classifying the entire image at once, this detector operates on $9 \times 9$ patches. The architecture is based on ResNet-50 (He et al., 2016) but uses $1 \times 1$ convolutions to limit the receptive field. The scores of all patches are combined using average pooling.

**Chen et al. (2024a)** This approach is a training paradigm that can increase the generalizability of AIGI detectors, by leveraging diffusion models to reconstruct semantically similar images containing detectable artifacts. The authors provide pre-trained detectors based on two backbones, ConvNeXt (Liu et al., 2022) and CLIP (Radford et al., 2021).

As we show in Section 3.2, the published version of most detectors does not perform well on the original images in our dataset. We hypothesize that this may be attributed to the applied postprocessing and the specific test images, which differ greatly from the training data of the various detectors. Since creating adversarial examples based on detectors that are already ineffective against clean samples reduces the impact of the work, we re-train detectors on the training subset described in Section 2.1, following the original authors' training instructions.

## 3 EXPERIMENTAL ANALYSIS

In this section, we evaluate how well RAID can be used to estimate the adversarial robustness of AIGI detectors. To this end, we initially test the performance on unperturbed images and conduct classical white-box attacks on each detector, demonstrating their susceptibility to evasion attacks. We subsequently analyze the transferability of white-box attacks and compare them to our ensemble attack, showing the effectiveness of RAID.

### 3.1 EXPERIMENTAL SETUP

**Detectors.** In addition to the seven detectors described in Section 2.2, we use four additional detectors whose architecture is a pre-trained visual backbone with a linear layer added on top. During training, a binary cross entropy loss is considered to discern between real and fake images. All model weights are frozen except for the linear layer trained using the $D^3$ train set. In particular, we use two different versions of DINO (Caron et al., 2021; Darcet et al., 2024) to highlight the behavior of self-supervised models. At the same time, to explore the impact of model size, we also consider a ViT-Tiny. We follow the transformation pipeline introduced by Ojha et al. (2023) for training and evaluation. This setup enables a consistent comparison across different architectures and training paradigms applied to deepfake detection. Finally, we adapt the CoDE model (Baraldi et al., 2024) for this analysis. Specifically, we used the feature extractor trained with a contrastive loss tailored

for deepfake detection and trained a linear classification layer on top. In this case, we follow the transformation strategy described in the original CoDE implementation.

**Dataset.** As noted in Section 2.2, we use the images of the $D^3$ test data set and the adversarial images generated by the adversarial attacks. We run the evaluations for our experiments on 1k clean images and 1k adversarial images. *All images are center-cropped* to ensure consistent input dimensions for efficient batch processing, before applying the detector-specific pre-processing and evaluation. Moreover, in the creation of the RAID dataset, following the approach used for the $D^3$ dataset, only images labeled as *safe* in the LAION metadata were considered as real images. This ensured that the generated images also adhere to this safeguard. To this end, samples depicting NSFW images has been excluded.

**Experimental Environment.** During the re-training phase of the different detectors, 4 A100 GPUs are used in a distributed data parallel setup. Each experiment runs for a maximum of 18 hours until convergence is reached. In contrast, the ensemble attack is conducted using a single A100 GPU, which takes a total of 8 hours. The evaluation script is lightweight and takes less than 1 hour for each detector in a non-distributed setting.

**Attack Setting.** The attack optimization is performed with the Projected Gradient Descent (PGD) adversarial attack (Madry et al., 2018)[1], a well-established iterative approach to generate adversarial examples. We employ PGD with 10 iterations and a step size of 0.05, and select three perturbation budgets for the attacks: $\epsilon = \frac{8}{255}$, $\epsilon = \frac{16}{255}$, and $\epsilon = \frac{32}{255}$ [2].

**Evaluation Metrics.** The performance of considered detectors is evaluated in terms of F1-score, accuracy, and AUROC. In particular, the **F1-score** measures the harmonic mean of the precision and true positive rate, which provides a metric capable of reliably computing the model performance in the presence of unbalanced class distributions. **Accuracy** is the ratio of correctly predicted samples over the total number of samples. It can be misleading in unbalanced datasets as it does not consider class distributions. We take the classification threshold equal to 0.5, as was done for all the detectors considered. Finally, **AUROC** summarizes the ROC curve, which plots the true positive rate against the false positive rate, by correctly measuring the capability of the model to identify samples across all classification thresholds. An AUROC of 1 corresponds to a perfect classifier with a 0 false positive rate across all thresholds, and an AUROC of 0.5 corresponds to a random chance classifier.

## 3.2 Experimental Results

**Initial Evaluation.** Prior to evaluating the adversarial robustness of the considered detectors, we first evaluate their performance on the $D^3$ test set. The initial performance evaluation of the detectors with the provided weights in Table 1 shows mixed results, particularly regarding F1-score and accuracy measures. For instance, Cavia et al. (2024) and Wang et al. (2020) show very low F1-score of 0.20 and 0.21, along with an AUROC of 0.45 and 0.46, respectively. Additionally, Ojha et al. (2023) shares a similar trend with F1-score equal to 0.33, while other detectors perform reasonably well. We hypothesize that the poor adaptability of the first three detectors is due to the data drift between the $D^3$ test set and the datasets used to training them, in addition to the post-processing applied to the images, especially the compression of images when using lossy formats.

Table 1: Evaluation of each model on a subset of the clean $D^3$ test set (1,000 samples). † refers to detectors trained on $D^3$.

| | F1 | Acc | AUROC |
|---|---|---|---|
| Cavia et al. (2024) (Ca24) | 0.20 | 0.07 | 0.45 |
| Corvi et al. (2023) (Co23) | 0.75 | 0.81 | 0.91 |
| Ojha et al. (2023) (O23) | 0.33 | 0.29 | 0.68 |
| Wang et al. (2020) (W20) | 0.21 | 0.02 | 0.46 |
| Cavia et al. (2024)† (Ca24†) | 0.55 | 0.63 | 0.84 |
| Chen et al. (2024a) (CLIP) (Ch24C) | 0.81 | 0.89 | 0.73 |
| Chen et al. (2024a) (ConvNext) (Ch24CN) | 0.86 | 0.91 | 0.94 |
| Corvi et al. (2023)† (Co23†) | 0.98 | 0.99 | 0.99 |
| Koutlis & Papadopoulos (2024) (K24) | 0.74 | 0.81 | 0.88 |
| Ojha et al. (2023)† (O23†) | 0.92 | 0.95 | 0.95 |
| Wang et al. (2020)† (W20†) | 0.99 | 0.99 | 0.99 |

---

[1] See Appendix A for a discussion of this choice and additional results with alternative attacks.

[2] The results for $\epsilon = \frac{8}{255}$ perturbation are reported in Appendix B.

Table 2: White-box and black-box adversarial robustness evaluation (F1-score/AUROC) on PGD with $\epsilon = 16/255$ perturbation. Adversarial attacks are ran against the model in each row and evaluated on models in each column. † refers to detectors trained on the $D^3$ training set. Final row $N- model$ refers to ensemble attacks targeting all models, excluding the evaluated against model. **Bold** values corresponding to white-box evaluation in single detector attacks.

| | Ca24† | Ch24C | Ch24CN | Co23† | K24 | O23† | W20† |
|---|---|---|---|---|---|---|---|
| Ca24† | **0.00/0.50** | 0.90/0.63 | 0.90/0.85 | 0.92/0.91 | 0.85/0.65 | 0.78/0.80 | 0.80/0.83 |
| Ch24C | 0.60/0.56 | **0.00/0.50** | 0.28/0.58 | 0.87/0.87 | 0.07/0.51 | 0.23/0.56 | 0.76/0.81 |
| Ch24CN | 0.69/0.54 | 0.86/0.61 | **0.29/0.48** | 0.87/0.88 | 0.85/0.59 | 0.79/0.80 | 0.75/0.80 |
| Co23† | 0.74/0.48 | 0.86/0.64 | 0.48/0.64 | **0.00/0.50** | 0.78/0.60 | 0.67/0.73 | 0.05/0.51 |
| K24 | 0.55/0.56 | 0.31/0.51 | 0.30/0.59 | 0.83/0.85 | **0.00/0.50** | 0.41/0.62 | 0.80/0.83 |
| O23† | 0.59/0.58 | 0.46/0.52 | 0.41/0.63 | 0.84/0.85 | 0.31/0.54 | **0.00/0.50** | 0.74/0.79 |
| W20† | 0.70/0.54 | 0.88/0.66 | 0.25/0.56 | 0.09/0.52 | 0.84/0.68 | 0.69/0.75 | **0.00/0.50** |
| $N- model$ | 0.66/0.43 | 0.34/0.51 | 0.20/0.56 | 0.53/0.67 | 0.08/0.51 | 0.18/0.54 | 0.40/0.62 |

Table 3: White-box and black-box adversarial robustness evaluation (F1-score/AUROC) on PGD with $\epsilon = 32/255$ perturbation. Adversarial attacks are ran against the model in each row and evaluated on models in each column. † refers to detectors trained on the $D^3$ training set. Final row $N- model$ refers to ensemble attacks targeting all models, excluding the evaluated against model. **Bold** values corresponding to white-box evaluation in single detector attacks.

| | Ca24† | Ch24C | Ch24CN | Co23† | K24 | O23† | W20† |
|---|---|---|---|---|---|---|---|
| Ca24† | **0.00/0.50** | 0.87/0.61 | 0.82/0.81 | 0.93/0.89 | 0.84/0.69 | 0.80/0.80 | 0.69/0.76 |
| Ch24C | 0.12/0.49 | **0.00/0.50** | 0.03/0.51 | 0.86/0.87 | 0.00/0.50 | 0.06/0.51 | 0.61/0.72 |
| Ch24CN | 0.33/0.50 | 0.79/0.61 | **0.25/0.51** | 0.81/0.83 | 0.79/0.69 | 0.76/0.76 | 0.58/0.70 |
| Co23† | 0.67/0.48 | 0.76/0.59 | 0.17/0.53 | **0.00/0.50** | 0.65/0.60 | 0.69/0.73 | 0.03/0.51 |
| K24 | 0.23/0.54 | 0.20/0.51 | 0.13/0.53 | 0.87/0.87 | **0.00/0.50** | 0.36/0.60 | 0.74/0.79 |
| O23† | 0.13/0.51 | 0.19/0.51 | 0.10/0.52 | 0.85/0.86 | 0.08/0.51 | **0.00/0.50** | 0.63/0.73 |
| W20† | 0.59/0.49 | 0.82/0.60 | 0.06/0.50 | 0.04/0.51 | 0.68/0.6 | 0.67/0.72 | **0.00/0.50** |
| $N- model$ | 0.28/0.33 | 0.08/0.49 | 0.01/0.50 | 0.32/0.59 | 0.01/0.50 | 0.06/0.52 | 0.17/0.54 |

In our work, we focus on robustness to adversarial attacks, and as such, we re-train on the $D^3$ training set the top four detectors that perform the worst: Cavia et al. (2024), Corvi et al. (2023), Ojha et al. (2023), and Wang et al. (2020). We retain the same weights for the rest of the detectors (Koutlis & Papadopoulos (2024), Chen et al. (2024a) (ConvNext), and Chen et al. (2024a) (CLIP)), both for the acceptable performance and to ensure the generalization of attacks without the induced risk of a dataset-bias if all detectors are trained on the $D^3$ dataset. After re-training, we note an improvement across metrics for the four detectors, except for Cavia et al. (2024) l which shows limited gains.

**Adversarial Robustness Evaluation.** We evaluate the adversarial robustness of the seven detectors, using the PGD attach with two perturbation budgets (*i.e.*, $\epsilon = \frac{16}{255}$ and $\epsilon = \frac{32}{255}$). Results are reported in Table 2 and Table 3, respectively. First, we assess the adversarial robustness of detectors in a white-box setting where each detector is attacked using adversarial perturbations crafted against it. This scenario represents the worst-case scenario in which the attacker has full knowledge of the target detector's architecture and parameters. The results for both $\epsilon = \frac{16}{255}$ and $\epsilon = \frac{32}{255}$ reveal a general lack of adversarial robustness. In particular, looking at Table 2, the F1-score drops to 0 for all detectors except for Chen et al. (2024a) (ConvNeXt), which exhibits inherent robustness with F1-score equal to 0.29. This tendency is carried over to the results in Table 3, where we report the attack against the detectors with $\epsilon = \frac{32}{255}$, as even under such large adversarial perturbations, the F1-score (0.25) does not drop to 0 similarly to the rest of the detectors. However, we underline that the drop in performance is still steep, from the initial F1-score of 0.87 on the clean examples reported in Table 1. Additionally, an evaluation of the AUROC reveals similar results.

Furthermore, we investigate the transferability of adversarial examples, referring to whether adversarial perturbations designed to fool one detector can also fool the others. Many adversarial ex-

Table 4: Evaluation of our attack methodology on AIGIs from additional generators. For each detector, we report F1-score, accuracy, and AUROC averaged over images from 10 generative models.

| Metric | Perturbation | Ca24[†] | Ch24C | Ch24CN | Co23[†] | K24 | O23[†] | W20[†] |
|---|---|---|---|---|---|---|---|---|
| **F1-score** | Clean | 0.41 | 0.91 | 0.86 | 0.95 | 0.96 | 0.74 | 0.91 |
| | 8/255 | 0.18 | 0.21 | 0.13 | 0.08 | 0.17 | 0.45 | 0.04 |
| | 16/255 | 0.00 | 0.00 | 0.15 | 0.00 | 0.01 | 0.05 | 0.00 |
| | 32/255 | 0.00 | 0.00 | 0.25 | 0.00 | 0.00 | 0.00 | 0.00 |
| **Accuracy** | Clean | 0.32 | 0.84 | 0.92 | 0.91 | 0.93 | 0.62 | 0.85 |
| | 8/255 | 0.18 | 0.19 | 0.15 | 0.13 | 0.17 | 0.34 | 0.11 |
| | 16/255 | 0.09 | 0.09 | 0.15 | 0.09 | 0.10 | 0.11 | 0.09 |
| | 32/255 | 0.09 | 0.09 | 0.21 | 0.09 | 0.09 | 0.09 | 0.09 |
| **AUROC** | Clean | 0.58 | 0.60 | 0.85 | 0.92 | 0.82 | 0.74 | 0.91 |
| | 8/255 | 0.53 | 0.54 | 0.52 | 0.52 | 0.54 | 0.56 | 0.51 |
| | 16/255 | 0.50 | 0.50 | 0.44 | 0.50 | 0.50 | 0.51 | 0.50 |
| | 32/255 | 0.01 | 0.01 | 0.50 | 0.01 | 0.01 | 0.01 | 0.01 |

amples generated for one detector do indeed reduce the performance substantially among the other detectors. For example, when we look at adversarial examples generated for Koutlis & Papadopoulos (2024) in Table 2, an AUROC performance drop can be seen across the following detectors: Cavia et al. (2024) to 0.56 (from 0.84), Chen et al. (2024a) (CLIP) to 0.51 (from 0.73), Chen et al. (2024a) (ConvNext) to 0.59 (from 0.94), and Ojha et al. (2023) to 0.5 (from 0.95). The other two detectors Corvi et al. (2023) and Wang et al. (2020) seem resilient, although this trend continues for perturbations generated for other detectors.

Next, we evaluate the transferability of ensemble attacks, in a *leave-one-out manner*, in which the assessed detector is excluded from the attacked ensemble. This ensemble approach, described in Section 2.2, significantly boosts transferability. It allows us to evaluate the detector's performance in a transferable black-box scenario where access to the targeted detector architecture or parameters is unavailable. Therefore, the attack (in a white-box manner) is done against an ensemble of detectors to increase its effectiveness further. We find that the ensemble attacks can drop the performance to metrics similar to a white-box attack. For example, looking at Table 2 (last row), we note that even for the two detectors Corvi et al. (2023) and Wang et al. (2020) that showed a decent robustness to perturbations generated for other detectors, the AUROC metric drops to 0.67 and 0.62 respectively (from 0.99 and 0.99), and the F1-score metric drops to 0.53 and 0.40 (from 0.98 and 0.99). This drop in performance is further highlighted in the last row of Table 3, where we use a higher perturbation budget in which the AUROC metric drops to 0.59 and 0.54 for the two detectors, respectively.

These results motivate the creation of our RAID dataset, which is composed of adversarial examples crafted using attacks on an ensemble of state-of-the-art detectors. Our benchmark dataset enables a fast and standardized evaluation of new detectors against strong transferable perturbations, facilitating the assessment of their robustness to adversarial attacks.

**Generalizability to AIGIs from Other Generators.** To demonstrate the adaptability of our method and broaden the evaluation scope, we apply our attack methodology to a set of 10 state-of-the-art text-to-image models (including SD3 (Esser et al., 2024), Kandinsky 2.1, Kandinsky 2.2, Kandinsky 3 (Razzhigaev et al., 2023), aMUSEd (Patil et al., 2024), PixArt-$\alpha$ (Chen et al., 2023b), SD-XL-Turbo (Podell et al., 2023), SAG (Hong et al., 2023), unCLIP (Ramesh et al., 2022), and FLUX.1-dev (Labs, 2024)). The selected models span a diverse range of architectures to test the generalizability of our approach under different conditions. We generate synthetic counterparts for the 4,800 real images included in our benchmark and apply our ensemble attack to create adversarial examples with varying perturbation budgets. In Table 4, we report F1-score, Accuracy, and AUROC, averaged over all 10 generators. While most detectors achieve decent results on clean images, their performance is drastically reduced on perturbed images. These results confirm the performance trends observed in Table 2 and Table 3, further demonstrating that our proposed approach is effective across multiple generator families and thus extends its applicability.

**RAID Benchmark: Tensors vs. Images.** Finally, we construct the RAID dataset as shown in Figure 2 by running the adversarial attack on the ensemble of detectors using the entire D$^3$ dataset. We generate adversarial examples and save them as PNG images, avoiding the use of lossy formats

Table 5: Adversarial robustness evaluation of the four trained baseline classifiers. All detectors are trained on the $D^3$ training set. RAID refers to the dataset of the generated adversarial examples saved as raw tensors, while RAID Img refers to the released dataset of adversarial images.

| Dataset | Perturbation | DINOv2 | | | DINOv2-Reg | | | ViT-T | | | ViT-T CoDE | | |
|---------|--------------|--------|-----|-------|------------|-----|-------|-------|-----|-------|------------|-----|-------|
| | | F1 | Acc | AUROC | F1 | Acc | AUROC | F1 | Acc | AUROC | F1 | Acc | AUROC |
| $D^3$ | Clean | 0.89 | 0.83 | 0.81 | 0.88 | 0.82 | 0.82 | 0.89 | 0.84 | 0.80 | 0.95 | 0.92 | 0.88 |
| RAID | 8/255 | 0.71 | 0.63 | 0.73 | 0.69 | 0.62 | 0.73 | 0.81 | 0.73 | 0.77 | 0.81 | 0.74 | 0.81 |
| RAID | 16/255 | 0.61 | 0.55 | 0.70 | 0.62 | 0.55 | 0.70 | 0.76 | 0.68 | 0.75 | 0.83 | 0.76 | 0.79 |
| RAID | 32/255 | 0.46 | 0.44 | 0.64 | 0.51 | 0.47 | 0.66 | 0.71 | 0.63 | 0.73 | 0.84 | 0.77 | 0.75 |
| RAID Img | 8/255 | 0.72 | 0.64 | 0.73 | 0.68 | 0.61 | 0.72 | 0.81 | 0.73 | 0.76 | 0.78 | 0.70 | 0.77 |
| RAID Img | 16/255 | 0.61 | 0.55 | 0.70 | 0.59 | 0.53 | 0.69 | 0.78 | 0.70 | 0.74 | 0.80 | 0.72 | 0.76 |
| RAID Img | 32/255 | 0.46 | 0.44 | 0.64 | 0.51 | 0.47 | 0.66 | 0.71 | 0.64 | 0.73 | 0.84 | 0.77 | 0.76 |

(*e.g.*, JPEG). The reason for this is that we only consider the worst-case adversarial scenario in which no post-processing operations are done, which could reduce the transferability of the attack. While the previous evaluation provides a good assessment of RAID, as only one detector is missing from the ensemble, we perform one additional evaluation on the full RAID dataset, to ensure that the effectiveness of the adversarial perturbations is not reduced with their quantization when we save them as images. We use four additional baseline detectors detailed in Section 3.1 tested against the adversarial examples saved as tensors with float values, and the adversarial images. We find no significant drop in effectiveness except for a few fluctuations as reported in Table 5. We release our benchmark with a total of 96,000 images: 24,000 adversarial examples for each attack parameter $\epsilon$ considering $\frac{8}{255}$, $\frac{16}{255}$ and $\frac{32}{255}$, in addition to the original images.

## 4 RELATED WORK

**Generative Image Modeling.** Learning to generate new samples from a high-dimensional data distribution, such as natural images, is not trivial. Several approaches, such as autoregressive models (van den Oord et al., 2016b;a), VAEs (Kingma & Welling, 2014; Rezende et al., 2014), and GANs (Goodfellow et al., 2014; Zhu et al., 2017; Choi et al., 2018; Karras et al., 2018; 2019; 2020; 2021), have been proposed. While it has been shown that DMs (Sohl-Dickstein et al., 2015; Ho et al., 2020; Dhariwal & Nichol, 2021) can surpass GANs with respect to quality, the costly iterative denoising process prevented the generation of high-resolution images. As a remedy, LDMs (Rombach et al., 2022) perform the diffusion and denoising process in the latent space of a pre-trained VAE (Kingma & Welling, 2014). Moreover, with the addition of cross-attention layers based on U-Net (Ronneberger et al., 2015), the generation can be controlled by textual prompts (Nichol et al., 2022; Ramesh et al., 2022; Saharia et al., 2022). Recent models have significantly advanced the resolution of generated images (up to 4k) (Chen et al., 2024b; Zhang et al., 2025), improved prompt following, and human preference (Esser et al., 2024; Xie et al., 2024).

**AIGI Detection Methods.** Most early approaches for detecint AIGIs exploit visible flaws like differently colored irises (Matern et al., 2019) or irregular pupil shapes (Guo et al., 2022). Since such imperfections are becoming less likely, several methods rely on imperceptible artifacts instead. Such features include model-specific fingerprints (Marra et al., 2019; Yu et al., 2019) or unnatural frequency patterns (Frank et al., 2020; Durall et al., 2020; Dzanic et al., 2020). Instead of using hand-crafted features, Wang et al. (2020) demonstrate that training a ResNet-50 (He et al., 2016) on real and generated images from ProGAN (Karras et al., 2018), paired with strong data augmentation, suffices to detect images generated by several other GANs. Subsequent works propose improved model architectures (Gragnaniello et al., 2021) or learning paradigms (Cozzolino et al., 2021; Mandelli et al., 2022; Chen et al., 2024a), with a particular focus on generalization (Chai et al., 2020; Liu et al., 2024). A promising direction is the use of foundation models as feature extractors to avoid overfitting on images generated by a single class of models (Ojha et al., 2023; Cozzolino et al., 2024; Koutlis & Papadopoulos, 2024). Recently, some works explore unique characteristics of DMs for detection, like frequency artifacts (Ricker et al., 2024b; Corvi et al., 2023) or features obtained by inverting the diffusion process (Wang et al., 2023; Ricker et al., 2024c; Cazenavette et al., 2024).

**Datasets for Evaluating the Robustness of AIGI Detectors.** To the best of our knowledge, we are the first to propose a dataset for evaluating the *adversarial* robustness of AIGI detectors. However, several datasets exist to test their generalizability and robustness to common image degradations. *GenImage* (Zhu et al., 2023) is a large-scale dataset comprising 1.35 million generated images based on ImageNet (Russakovsky et al., 2015). *WildFake* (Hong & Zhang, 2024) features images generated by GANs, DMs, and other generative models, which are partly sourced from platforms such as Civitai to cover a broad range of content and styles. Yan et al. (2025) also collect images from popular image-sharing websites. However, their *Chameleon* dataset features images that were misclassified by human annotators, making them particularly challenging. Furthermore, *Deepfake-Eval-2024* (Chandra et al., 2025) contains deepfake videos, audio, and images that circulated on social media and deepfake detection platforms in 2024.

**Adversarial Robustness of AIGI Detectors.** The vulnerability of deepfake detectors to adversarial examples was first explored by Carlini & Farid (2020). They demonstrate that by adding visually imperceptible perturbations to an image, the AUC of a forensic classifier can be reduced from 0.95 down to 0.0005 in the white-box and 0.22 in the black-box setting. Subsequent work explores the applicability of attacks in practical scenarios (Neekhara et al., 2021; Hussain et al., 2021; Mavali et al., 2024) as well as possible defenses (Gandhi & Jain, 2020). De Rosa et al. (2024) study the robustness of CLIP-based detectors, finding that adversarial examples computed for CNN-based classifiers are not easily transferable and vice versa. Besides the addition of adversarial noise, it has been shown that deepfake detectors can also be attacked by applying natural degradations (e.g., local brightness changes) (Hou et al., 2023) or by removing generator-specific artifacts (Dong et al., 2022; Wesselkamp et al., 2022). Other attacks leverage image generators themselves to perform semantic adversarial attacks, which adversarially manipulate a particular attribute of an image (Meng et al., 2024) that can even be controlled through a text prompt (Liu et al., 2023; Abdullah et al., 2024).

## 5 DISCUSSION

Adversarial robustness should always be evaluated when proposing new AIGI detection methods, as in current state-of-the-art detectors, it is vastly neglected in favor of an evaluation against *naturally occurring* post-processing operations, such as resizing, cropping, blurring, JPEG compression or noise. While the robustness to these operations is indeed important, introducing a *malicious actor* that utilizes carefully crafted adversarial noise can lead to the evasion of detection by most methods, as highlighted in our work. Nonetheless, the lack of a standard benchmark that serves as a comparability reference for detectors contributes further to this lack of evaluation against adversarial attacks in AIGI detection. As such, we introduce RAID to address this gap in the current literature and provide a more comprehensive solution for evaluating generative models. However, it is important to acknowledge that our method is not without its limitations. One key challenge is that RAID requires frequent updates to stay relevant as new generative models emerge, and these models would need to be incorporated into our proposed ensemble attacks for continued effectiveness. This dynamic nature of generative models demands a proactive approach to maintain the robustness of RAID over time. Additionally, our perturbations are not designed to be robust to post-processing operations, which should be considered in future work. Finally, due to the inherent restrictions of input sizes of the architecture of some detectors, when considering attacks on ensemble models, we are restricted to the center region of the image to be perturbed.

## 6 CONCLUSION

We introduce RAID, the first benchmark dataset of transferable adversarial examples for robustness evaluation of AIGI detection. We employ an ensemble attack that demonstrates strong transferability against seven diverse detectors and cover images generated from four text-to-image generative models. Our results further highlight the existing gap in current evaluations of state-of-the-art detectors, as more often than not, they are tested on naturally occurring post-processing as images are disseminated and shared, but remain highly vulnerable to adversarial attacks. RAID addresses this gap by providing a simple and reliable benchmark for adversarial robustness evaluation, ensuring that detection models can be tested under more realistic and challenging adversarial conditions.

ETHICS STATEMENT

Research on the adversarial robustness of classifiers always poses a dual-use risk. Our proposed attack methodology could be misused by an adversary to craft AIGIs that bypass detectors. However, our work does not introduce any new attack methodologies that were not previously available to adversaries. Therefore, we argue that our benchmark dataset primarily benefits defenders, enabling developers to evaluate and improve the robustness of their models.

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

## A  OTHER ADVERSARIAL ATTACKS: DISCUSSION AND RESULTS

To complement our PGD-based ensemble attack, we conduct additional experiments comparing PGD with two widely used alternatives: APGD (Croce & Hein, 2020) and CWA (Chen et al., 2023a). The comparison is carried out on a subset of 1,000 images with perturbation budget $\epsilon = 32/255$, and results, evaluated in terms of F1-score and AUROC, are reported in Table 6.

In particular, APGD is the only component of the AutoAttack suite that can be extended to an ensemble by averaging per-model cross-entropy losses. However, our results indicate that this adaptation is ineffective in the transfer setting. The attack is designed for single-model optimization and, when applied to multiple models simultaneously, its internal heuristics (*e.g.*, adaptive step-size, frequent restarts) are destabilized by conflicting gradient signals. As a result, APGD yields significantly higher F1-score and AUROC values than PGD, indicating weaker adversarial examples. CWA, on the other hand, is explicitly designed for transferability and reports performance closer to PGD across detectors. Nevertheless, our ensemble PGD is the only method that consistently drives detector performance down toward chance level, especially in terms of AUROC (*i.e.*, often close to 0.50) and F1-score (*i.e.*, often close to 0.20). This demonstrates that our attack is both effective and stable when optimized against a diverse set of detectors.

As shown, while CWA provides a competitive baseline for transfer-based attacks, our ensemble-targeted PGD achieves the most reliable reduction of robustness across models. This supports our choice of PGD as the basis for RAID, where the goal is to release a pre-computed set of transferable adversarial examples that are both practical and sufficiently strong to serve as a benchmark.

## B  ADDITIONAL ADVERSARIAL ROBUSTNESS EVALUATIONS

**Sensitivity to Image Transformations.** In our experiments, the attack aims to provide an approximate estimation of the adversarial robustness in worst-case settings. For this reason, we do not consider optimizing the adversarial perturbation to make it robust against any transformations. Nevertheless, this could be achieved by introducing data augmentations during the attack optimization and leveraging the Expectation Over Transformation (EOT) algorithm (Athalye et al., 2018), although we expect that optimizing against transformations would decrease the adversarial example transferability on other never-before-seen detectors. Likely, in real scenarios, an attacker would perform a stronger, yet more costly, attack tailored to the target detector that may resist image transformations better. In this context, our dataset can quickly and costless provide valuable insights: if a model is not robust with respect to it, it will almost certainly be vulnerable to more realistic threat models (Carlini et al., 2019).

Nonetheless, to provide additional insights, we evaluate the models on a subset of 1,000 samples from our dataset with $\epsilon = 32/255$ by applying several transformations before classifying the images. The results in Table 7 show that, while reduced with respect to when no transformations are applied, the attack still shows a considerable effectiveness in fooling the detectors.

**Results using $\epsilon = 8/255$ Perturbation.** For completeness, we also evaluate the detectors under a smaller perturbation budget of $\epsilon = 8/255$. This setting is commonly used in white-box evaluations but is less relevant in the transfer-based scenario, where larger perturbations (*e.g.*, $\epsilon = 16/255$ or $\epsilon = 32/255$) are typically required to achieve effective cross-model fooling. The results reported in Table 8 confirm this intuition. In the white-box case, where the attack directly targets a single detector, performance drops markedly, with F1-scores approaching zero and AUROC close to 0.50.

Table 6: Comparison with other attacks at $\epsilon = 32/255$.

| Metric | Attack | Ca24[†] | Ch24C | Ch24CN | Co23[†] | K24 | O23[†] | W20[†] |
|---|---|---|---|---|---|---|---|---|
| **F1-score** | PGD (Ours) | 0.28 | 0.08 | 0.01 | 0.32 | 0.01 | 0.06 | 0.17 |
| | APGD | 0.43 | 0.92 | 0.41 | 0.74 | 0.80 | 0.90 | 0.93 |
| | CWA | 0.07 | 0.40 | 0.17 | 0.23 | 0.34 | 0.00 | 0.19 |
| **AUROC** | PGD (Ours) | 0.33 | 0.49 | 0.50 | 0.59 | 0.50 | 0.52 | 0.54 |
| | APGD | 0.40 | 0.80 | 0.63 | 0.79 | 0.82 | 0.90 | 0.93 |
| | CWA | 0.32 | 0.25 | 0.50 | 0.34 | 0.31 | 0.05 | 0.25 |

Table 7: Performance metrics under different transformations. Results are reported in the $N-model$ setting, using PGD with $\epsilon = 32/255$.

| Metric | Setting | Ca24[†] | Ch24C | Ch24CN | Co23[†] | K24 | O23[†] | W20[†] |
|---|---|---|---|---|---|---|---|---|
| **F1-score** | No transform | 0.28 | 0.08 | 0.01 | 0.32 | 0.01 | 0.06 | 0.17 |
| | Resize 64×64 | 0.35 | 0.86 | 0.35 | 0.40 | 0.10 | 0.01 | 0.45 |
| | Random Crop 64×64 | 0.49 | 0.87 | 0.02 | 0.60 | 0.01 | 0.01 | 0.61 |
| | Random Flip | 0.25 | 0.21 | 0.01 | 0.32 | 0.25 | 0.05 | 0.13 |
| | JPEG Compression | 0.03 | 0.18 | 0.03 | 0.18 | 0.10 | 0.09 | 0.11 |
| **Accuracy** | No transform | 0.24 | 0.22 | 0.20 | 0.34 | 0.22 | 0.20 | 0.27 |
| | Resize 64×64 | 0.34 | 0.77 | 0.35 | 0.40 | 0.24 | 0.20 | 0.42 |
| | Random Crop 64×64 | 0.40 | 0.78 | 0.20 | 0.53 | 0.20 | 0.20 | 0.54 |
| | Random Flip | 0.22 | 0.27 | 0.20 | 0.35 | 0.31 | 0.22 | 0.25 |
| | JPEG Compression | 0.17 | 0.26 | 0.20 | 0.27 | 0.24 | 0.23 | 0.25 |
| **AUROC** | No transform | 0.33 | 0.49 | 0.50 | 0.59 | 0.50 | 0.52 | 0.54 |
| | Resize 64×64 | 0.53 | 0.56 | 0.56 | 0.61 | 0.52 | 0.50 | 0.61 |
| | Random Crop 64×64 | 0.47 | 0.52 | 0.50 | 0.65 | 0.50 | 0.49 | 0.65 |
| | Random Flip | 0.32 | 0.49 | 0.50 | 0.58 | 0.56 | 0.51 | 0.53 |
| | JPEG Compression | 0.41 | 0.51 | 0.50 | 0.54 | 0.52 | 0.51 | 0.53 |

Table 8: White-box and black-box adversarial robustness evaluation (F1-score/AUROC) on PGD with $\epsilon = 8/255$ perturbation. Adversarial attacks are ran against the model in each row and evaluated on models in each column. † refers to detectors trained on the $D^3$ training set. Final row $N-model$ refers to ensemble attacks targeting all models, excluding the evaluated against model. **Bold** values corresponding to white-box evaluation in single detector attacks.

| | Ca24[†] | Ch24C | Ch24CN | Co23[†] | K24 | O23[†] | W20[†] |
|---|---|---|---|---|---|---|---|
| Ca24[†] | **0.00/0.50** | 0.89/0.59 | 0.90/0.87 | 0.91/0.91 | 0.88/0.59 | 0.78/0.80 | 0.87/0.88 |
| Ch24C | 0.64/0.59 | **0.71/0.50** | 0.70/0.75 | 0.86/0.87 | 0.50/0.60 | 0.48/0.64 | 0.83/0.86 |
| Ch24CN | 0.71/0.56 | 0.89/0.62 | **0.62/0.55** | 0.91/0.91 | 0.87/0.53 | 0.74/0.77 | 0.85/0.87 |
| Co23[†] | 0.71/0.50 | 0.89/0.64 | 0.81/0.80 | **0.00/0.50** | 0.86/0.56 | 0.67/0.74 | 0.30/0.58 |
| K24 | 0.63/0.60 | 0.57/0.55 | 0.62/0.72 | 0.85/0.87 | **0.09/0.51** | 0.44/0.62 | 0.84/0.86 |
| O23[†] | 0.65/0.61 | 0.76/0.58 | 0.77/0.79 | 0.84/0.86 | 0.64/0.59 | **0.00/0.50** | 0.82/0.85 |
| W20[†] | 0.70/0.56 | 0.90/0.62 | 0.69/0.73 | 0.27/0.57 | 0.87/0.59 | 0.67/0.74 | **0.00/0.50** |
| $N-model$ | 0.68/0.49 | 0.74/0.59 | 0.63/0.70 | 0.59/0.71 | 0.57/0.61 | 0.45/0.63 | 0.49/0.66 |

In contrast, in the black-box transfer setting, detectors maintain relatively high robustness, with substantially higher F1-score and AUROC values compared to the results reported in Table 2 and Table 3 for $\epsilon = 16/255$ and $\epsilon = 32/255$, respectively.

**Variance Across Test Splits.** We provide additional details regarding the variability of our experimental results due to different test splits. As such, we conducted the main experiments using five random seeds. The PGD attack is used with 10 steps, step size equal to 0.05, and $\epsilon = 16/255$. We report means and standard deviations in Table 9 and Table 10, respectively for F1-score and AUROC values. A similar trend can be observed where the ensemble attacks are a good estimate for the adversarial robustness of the AIGI detectors.

Table 9: White-box and black-box adversarial robustness evaluation (mean of the F1 scores across five random seeds $\pm$ standard deviation).

| | Ca24$^\dagger$ | Ch24C | Ch24CN | Co23$^\dagger$ | K24 | O23$^\dagger$ | W20$^\dagger$ |
|---|---|---|---|---|---|---|---|
| Ca24$^\dagger$ | **0.00±0.00** | 0.87±0.0 | 0.84±0.01 | 0.86±0.0 | 0.81±0.02 | 0.71±0.02 | 0.66±0.02 |
| Ch24C | 0.54±0.01 | **0.00±0.00** | 0.24±0.02 | 0.75±0.01 | 0.08±0.01 | 0.23±0.03 | 0.56±0.01 |
| Ch24CN | 0.61±0.01 | 0.83±0.01 | **0.26±0.02** | 0.73±0.01 | 0.81±0.01 | 0.73±0.01 | 0.56±0.01 |
| Co23$^\dagger$ | 0.70±0.00 | 0.82±0.00 | 0.44±0.01 | **0.00±0.00** | 0.75±0.02 | 0.63±0.02 | 0.02±0.01 |
| K24 | 0.49±0.02 | 0.23±0.03 | 0.26±0.02 | 0.71±0.01 | **0.07±0.02** | 0.35±0.02 | 0.60±0.02 |
| O23$^\dagger$ | 0.51±0.02 | 0.38±0.02 | 0.35±0.02 | 0.73±0.01 | 0.29±0.03 | **0.00±0.00** | 0.55±0.01 |
| W20$^\dagger$ | 0.64±0.01 | 0.84±0.01 | 0.21±0.02 | 0.08±0.01 | 0.79±0.01 | 0.63±0.01 | **0.00±0.00** |
| $N-model$ | 0.61±0.01 | 0.30±0.02 | 0.19±0.03 | 0.47±0.02 | 0.09±0.01 | 0.21±0.03 | 0.3±0.02 |

Table 10: White-box and black-box adversarial robustness evaluation (mean of the AUROC scores across five random seeds $\pm$ standard deviation).

| | Ca24$^\dagger$ | Ch24C | Ch24CN | Co23$^\dagger$ | K24 | O23$^\dagger$ | W20$^\dagger$ |
|---|---|---|---|---|---|---|---|
| Ca24$^\dagger$ | **0.50± 0.00** | 0.62±0.02 | 0.81±0.01 | 0.85±0.00 | 0.69±0.02 | 0.75±0.01 | 0.75±0.01 |
| Ch24C | 0.53±0.01 | **0.50±0.00** | 0.57±0.01 | 0.79±0.01 | 0.52±0.00 | 0.56±0.01 | 0.70±0.00 |
| Ch24CN | 0.51±0.02 | 0.62±0.02 | **0.49±0.01** | 0.78±0.00 | 0.61±0.02 | 0.74±0.01 | 0.70±0.01 |
| Co23$^\dagger$ | 0.48±0.02 | 0.62±0.02 | 0.61±0.01 | **0.50±0.00** | 0.63±0.01 | 0.7±0.01 | 0.51±0.00 |
| K24 | 0.53±0.01 | 0.51±0.01 | 0.57±0.00 | 0.77±0.00 | **0.51±0.01** | 0.59±0.01 | 0.72±0.01 |
| O23$^\dagger$ | 0.53±0.01 | 0.53±0.01 | 0.60±0.01 | 0.78±0.00 | 0.55±0.02 | **0.50±0.00** | 0.69±0.01 |
| W20$^\dagger$ | 0.51±0.01 | 0.65±0.01 | 0.53±0.01 | 0.52±0.00 | 0.66±0.01 | 0.71±0.01 | **0.50±0.00** |
| $N-model$ | 0.42±0.02 | 0.53±0.01 | 0.55±0.01 | 0.65±0.01 | 0.52±0.01 | 0.55±0.01 | 0.59±0.01 |

Table 11: Performance on commercial deepfake detectors. We report mean detector scores on AIGIs, with adversarial variants evaluated on RAID examples. Metrics on real images are also included.

| Detector | Score | Score (Real) | Adversarial Score | Acc | Acc (Real) | Adversarial Acc |
|---|---|---|---|---|---|---|
| Sightengine | 0.65 | 0.99 | 0.01 | 0.66 | 1.00 | 0.00 |
| HIVE | 0.59 | 0.99 | 0.45 | 0.60 | 1.00 | 0.44 |

## C   EVALUATION ON COMMERCIAL DETECTORS

We use two commercial AIGI detectors provided by Sightengine[3] and HIVE[4] to evaluate RAID in real-world settings. We report the results set of 50 real images and 50 pairs of clean and adversarial images in Table 11. While these results are preliminary due to the dataset size, they still suggest that RAID retains effectiveness when assessing the adversarial robustness of detectors deployed in commercial detection APIs. We compute the Accuracy with a 0.5 threshold on the confidence score reported by the detectors. We also provide a few examples of the detection scores returned by commercial detectors on randomly selected images in Figure 3 and Figure 4.

## D   LLM USAGE

In this paper, LLMs were used only for minor writing polish. They did not contribute to the design of experiments, the analysis of results, or the generation of scientific content.

---

[3]https://dashboard.sightengine.com/ai-image-detection
[4]https://hivemoderation.com/ai-generated-content-detection

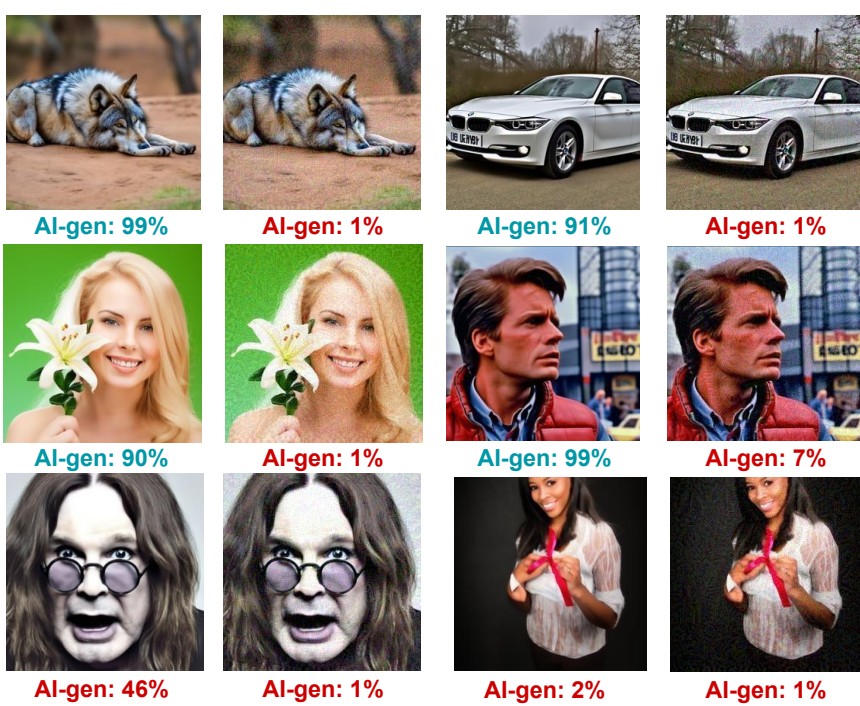

Figure 3: Detection scores returned by Sightengin detector on a subset of clean and adversarial AI-generated images. Higher scores indicate higher confidence that the image is AI-generated.

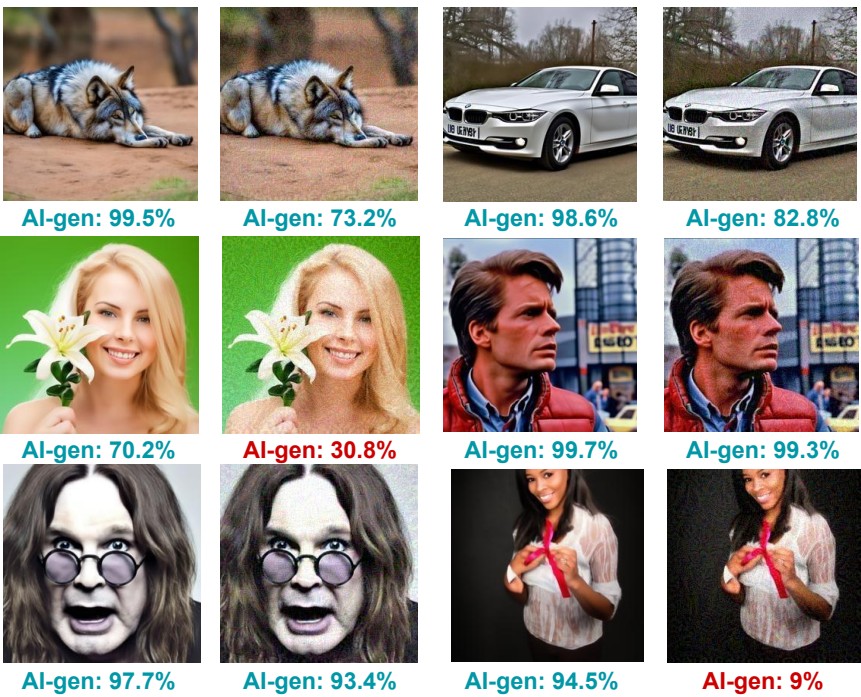

Figure 4: Detection scores returned by HIVE Moderation detector on a subset of clean and adversarial AI-generated images. Higher scores indicate higher confidence that the image is AI-generated.

