# OpenReview forum: "RAID: A Benchmark Dataset for Testing the Adversarial Robustness of AI-Generated Image Detectors"
_ICLR.cc/2026/Conference — ICLR 2026 Conference Withdrawn Submission_

### Official Review · Reviewer_vxxp · 2025-10-30

**Soundness:** 3
**Presentation:** 3
**Contribution:** 2
**Rating:** 2
**Confidence:** 5

**Summary:**

This paper provides a new benchmark on evaluating the Adversarial Robustness of AI-generated image Detectors based on ensemble transfer-based adversarial examples. A diverse set og text-to-image models and detection methods are considered.

**Strengths:**

1. The paper is well written, and the technical details are sufficient for basic understanding.
2. Multiple detectors and generative models are tested.
3. White-box scenarios are also presented for comparison.

**Weaknesses:**

1. No detectors from 2025 are considered, for both ensemble and evaluation. Considering that AIGI detection is quite a hot topic, it is necessary to test methods published recently.

2. Only PGD with 10 iterations and a step size of 0.05 is tested. Although APGD and CWA are discussed in the appendix, the experiments are not thorough. More importantly, only one transfer technique, CWA, is considered. This is insufficient because this paper focuses on transfer attack settings. The authors should further consider more representative transfer techniques, such as MI, DI, TI, and SI. Please see https://xiaosenwang.com/transfer_based_attack_papers.html for a curated list of transfer attack papers. It is also weird that CWA, which is explicitly designed for transferability, performs even worse than PGD.

3. No comparisons to existing attacks specifically designed for AIGI detection, especially those mentioned in Section 4 Related Work: Adversarial Robustness of AIGI Detectors.

4. The authors claim that “The adversarial robustness is often not investigated in works proposing synthetic image detectors.” The reviewer does not think this is a problem, but a reasonable common practice. This is just like every paper that proposes a new technique (e.g., model architecture, loss, etc) for improving the general performance in a specific task would not test the adversarial robustness. Adversarial robustness is a quite independent problem that just considers the ideal, worst-case scenarios.

5. The authors state that “each generated image is post-processed such that the image format and compression strength match that of the real distribution present in the corresponding real image.” Please provide more details about this. In particular, were the generated images originally generated with the same size or post-resized to the same size?

6. Fooling the detector for misprediction "from fake to real" and "from real to fake" are both considered. The former makes sense and was commonly studied in the literature because it allows bypassing the detector with a fake image. However, the real-world meaning of the latter should be discussed.

7. The first paragraph is too long. Starting the paper directly with the AIGI detection is enough. Instead, it makes more sense to use more content to introduce more details about RAID. Now, even the statistics of the dataset are not mentioned in the Introduction, making the significance of the contribution unclear. More specifically, it is better to have a table to show the experimental setting, e.g., the data splits and selection of generative models.

**Questions:**

See the above weaknesses.

---

### Official Review · Reviewer_fGJU · 2025-10-30

**Soundness:** 3
**Presentation:** 4
**Contribution:** 2
**Rating:** 6
**Confidence:** 4

**Summary:**

The paper addresses a practical but often overlooked issue in evaluating AI-generated image (AIGI) detectors: most current evaluations assume ideal, non-adversarial settings and lack a standard way to test robustness against attacks. To address this, the authors introduce RAID, a benchmark dataset built to measure how well AIGI detectors withstand adversarial perturbations.
RAID includes 96,000 images, among which 72,000 are adversarial examples generated using the PGD method with three perturbation levels (ε = 8/255, 16/255, 32/255). These are created from four popular text-to-image diffusion models (such as the Stable Diffusion family and DeepFloyd IF) and produced by attacking seven representative AIGI detectors with diverse architectures.
Experiments show that RAID’s adversarial samples transfer well to unseen detectors and can effectively reveal weaknesses in robustness. Even the best-performing models, such as the ConvNeXt-based detector from Chen et al. (2024a), suffer large drops in F1 score under attack. RAID also generalizes across ten additional generative models, four baseline detectors, and two commercial detectors (Sightengine and HIVE), demonstrating its broad applicability. While the main contribution lies in dataset and benchmark construction rather than new algorithms or theory, the work provides a valuable tool for systematic robustness evaluation.

**Strengths:**

1. The paper fills a clear gap in current research: there has been no standardized dataset for testing the adversarial robustness of AIGI detectors. RAID provides a reproducible and comparable benchmark that can guide future studies.
2. Well-designed dataset
(1) By combining seven different detector architectures (ResNet-based, CLIP-based, and patch-level models) during attack generation, the resulting adversarial samples are not tied to any single model, making them more general and closer to real-world black-box scenarios.
(2) The dataset covers 96,000 images from four major generators, carefully matching real-image compression and format to reduce bias. Three perturbation budgets allow for both imperceptible and stronger attacks, fitting different robustness evaluation settings.
(3) The dataset provides lightweight evaluation scripts, and images are stored in PNG format to avoid additional compression loss.
3. Comprehensive and careful experiments
(1) RAID is tested on a range of models, including extra baseline detectors and commercial systems, confirming its robustness and generality.
(2) The ensemble PGD approach is compared with other attacks such as APGD and CWA, showing stronger and more stable performance.
(3) Using F1, accuracy, and AUROC avoids bias from any single measure and provides a fuller view of robustness degradation.

**Weaknesses:**

1. The contribution is mainly engineering-focused. It builds on established methods like PGD and ensemble attack concepts without proposing new theoretical ideas or insights into the mechanisms behind detector robustness.
2. While the authors note that RAID should evolve with new generative models, they do not describe how future updates will be handled. The dataset relies on diffusion-based generators, which may limit its relevance once new architectures emerge. No clear plan for versioning or community maintenance is mentioned.
3. The adversarial examples are not optimized for robustness under image operations such as cropping or JPEG compression. Although the authors focus on “worst-case” scenarios, real-world AIGI often goes through such transformations, which can reduce attack effectiveness.
4. Missing details:
(1)The results for smaller perturbations (ε = 8/255) are only briefly discussed and not well analyzed, leaving the reasons for weaker transfer unexplored.
(2)The evaluation on commercial detectors is based on a very small sample (50 real and 50 generated images per model), which makes the results less reliable.
5. The study does not suggest any possible defense methods or insights drawn from RAID. Even a short discussion on how RAID’s findings might inspire robustness improvements would add value.

**Questions:**

1. How was the “matching compression level” between generated and real images determined? Was it derived from LAION metadata or measured automatically? The criteria for selecting the 4,800 new real images (e.g., scene diversity, resolution) should also be clarified.
2. What is the rationale for using 10 PGD iterations and a 0.05 step size? Was this combination empirically optimized or simply adopted from prior work?
3. Visual perceptibility of adversarial samples: The paper relies solely on ε values to claim imperceptibility, but does not conduct human perceptual validation (e.g., user studies). If ε = 32/255 produces visible noise, its practicality as a realistic adversarial example is questionable.
4. The authors describe RAID as “engineering infrastructure,” but do they plan to extract theoretical insights from RAID’s results (e.g., correlations between robustness and architecture design)? Clarifying this could enhance the paper’s academic contribution.

---

### Official Review · Reviewer_jmcu · 2025-10-30

**Soundness:** 2
**Presentation:** 3
**Contribution:** 2
**Rating:** 2
**Confidence:** 2

**Summary:**

This paper introduces RAID, a large benchmark of precomputed adversarial examples for AI-generated image detectors. RAID is produced by attacking an ensemble of seven detectors on images synthesized by multiple text-to-image models and saved at three perturbation levels. The authors claim these adversarial examples are highly transferable and that RAID can serve as a fast standardized test for a detector’s adversarial robustness. They provide white-box, black-box, and ensemble attack experiments and proposed benchmark dataset
consists of 72,000 adversarial examples – 24,000 adversarial examples for each attack parameter $\epsilon(\frac{8}{255}, \frac{16}{255}, \frac{32}{255})$ – in addition to original images, for a total of 96,000 images.

**Strengths:**

1. Evaluating adversarial robustness of AIGI detectors is timely and societally relevant as image generation quality improves; providing a standardized benchmark addresses a real community need.

2. The authors evaluate multiple published detectors and multiple generative models, improving the generality of the findings.

3. The dataset consists 96k files which is a substantial engineering contribution that will help the researchers to test detectors.

**Weaknesses:**

1. The authors used cross-entropy loss toward a target class (Eq. (1)–(3)) but the text does not clearly state whether attacks are targeted or untargeted in practice nor how the target label is chosen. This is extremely important as targeted and untargeted attacks have different success/transferability properties and different real-world relevance. The threat model needs a precise, explicit statement.

2. The paper re-trains weaker detectors on the D3 training set while leaving three detectors with original weights. The stated reason is to avoid creating adversarial examples from detectors that already perform poorly on D3. However, re-training detectors on the same D3 distribution which is also used to craft the attacks and to build RAID risks data / attack circularity and may bias the benchmark toward vulnerabilities specific to that dataset/ensemble. The mix of re-trained and original weights also complicates interpreting transferability. If ensemble and attack generation are closely related to D3 then RAID might overestimate transferability to detectors trained on different datasets or real deployed detectors.

3. The adversarial perturbations are saved as PNG tensors (lossless) and the authors explicitly do not optimize for robustness to real post-processing (JPEG, rescaling, social-media recompression). They acknowledge this limitation but nevertheless claim RAID is a reliable estimate of robustness. Generally, AIGIs are shared through pipelines. Attacks that fail after common transformations (or that become visible/obvious) are less relevant in practice. The authors show some transformation experiments in appendix but leave out important real pipelines

4. The authors use ensemble PGD as their main attack and compare adaptions of APGD and CWA. However, the adversarial robustness literature emphasizes diverse, adaptive, and strong attacks plus gradient-free approaches for transferability. Some of these are only briefly discussed in Appendix and the APGD adaptation is argued to be ineffective without deeper diagnostics. A benchmark claiming to reflect “strong transferable attacks” should show RAID withstands other strong attack classes.

5. The paper reports single number digits in the main tables. They have mentioned variance over 5 seeds in the appendix only for a few tables and the main text mostly uses point estimation. For an empirical dataset/benchmark, error bars are necessary.

**Questions:**

See the weaknesses.

**Details Of Ethics Concerns:**

The ethics statement acknowledges dual-use but largely dismisses the danger by stating 'our work does not introduce any new attack methodologies'. This is insufficient according to me because publishing a large scale, easy to use attack corpus lowers the bar for malicious acotrs.

---

### Note · Authors · 2025-11-21

I have read and agree with the venue's withdrawal policy on behalf of myself and my co-authors.